# FedNAS: Federated Deep Learning via Neural Architecture Search

## Abstract

Federated Learning (FL) is an effective learning framework used when data cannot be centralized due to privacy, communication costs, and regulatory restrictions. While there have been many algorithmic advances in FL, significantly less effort has been made on model development, and most works in FL employ predefined model architectures discovered in the centralized environment. However, these predefined architectures may not be the optimal choice for the FL setting since the user data distribution at FL users is often non-identical and independent distribution (non-IID). This well-known challenge in FL has often been studied at the optimization layer. Instead, we advocate for a different (and complementary) approach. We propose Federated Neural Architecture Search (FedNAS) for automating the model design process in FL. More specifically, FedNAS enables scattered workers to search for a better architecture in a collaborative fashion to achieve higher accuracy. Beyond automating and improving FL model design, FedNAS also provides a new paradigm for personalized FL via customizing not only the model weights but also the neural architecture of each user. As such, we also compare FedNAS with representative personalized FL methods, including perFedAvg (based on meta-learning), Ditto (bi-level optimization), and local fine-tuning. Our experiments on a non-IID dataset show that the architecture searched by FedNAS can outperform the manually predefined architecture as well as existing personalized FL methods. To facilitate further research and real-world deployment, we also build a realistic distributed training system for FedNAS, which will be publicly available and maintained regularly.

## 1 Introduction

Federated Learning (FL) is a promising approach for decentralized machine learning, which aims to avoid data sharing and lower the communication cost (McMahan et al., 2016). As such, it has gained a lot of attention in various domains of machine learning such as computer vision, natural language processing, and data mining. Despite its widespread popularity, one of the key challenges of FL is data heterogeneity. Since users' data are not identically or independently distributed (non-IID) in nature, a globally learned model may not perform optimally on all user devices. When interweaving with data heterogeneity, data invisibility is another issue that has rarely been studied. For this reason, to find a better model architecture with higher accuracy, developers must design or choose multiple architectures, then tune hyperparameters remotely to fit the scattered data. This process is extremely expensive because attempting many rounds of training on edge devices results in a remarkably higher communication cost and on-device computational burden than the data center environment.

To mitigate the challenge of data heterogeneity, researchers have proposed methods to train a global model, including `FedProx` Li et al. (2018), `FedNova` Wang et al. (2020), and `FedOPT` Reddi et al. (2020). Additionally, personalized frameworks such as `Ditto` Li et al. (2020), `pFedMe` Dinh et al. (2020), and `PerFedAvg` Fallah et al. (2020) have been recently developed to optimize personalized models to adapt to individual user's data. These prior works have made remarkable progress in designing optimization schemes for pre-defined model architectures operated at pure optimization. However, these algorithms all require lots of effort to tune hyperparameters; this is attributed to their strong prior assumptions, which may not always match the unknown data distribution. For example, practitioners must tune the regularization parameter in Ditto Li et al. (2020) and pFedMe Dinh et al. (2020) to find a proper correlation between the aggregated global model and local model. Moreover,

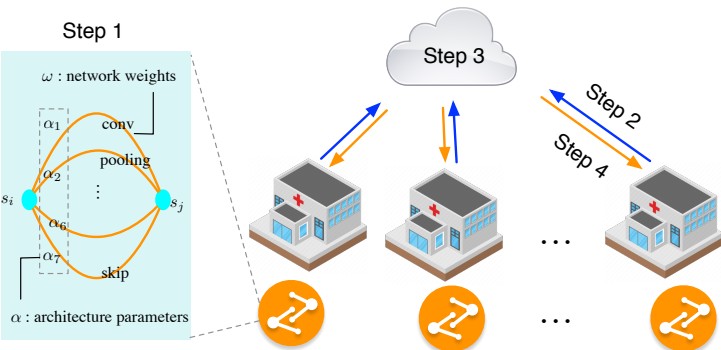

Figure 1: Illustration of Federated Neural Architecture Search (*step 1*: search locally; *step 2*: send the gradients of $\alpha$ and $w$ to the server; *step 3*: merge gradients to get global $\alpha$ and $w$; *step 4*: synchronize the updated $\alpha$ and $w$ to each client.)

their design is only in optimization level and does not consider the efficacy of model selection and neural architecture design, leading to a suboptimal solution when using a pre-defined model.

We aim to address data heterogeneity in FL via a different and complementary approach that is based on *model personalization through neural architecture search* (NAS). NAS has recently gained much momentum to adapt heterogeneity in neural architecture design Smithson et al. (2016); Chen et al. (2020), latency Wu et al. (2019); Tan et al. (2019); Cai et al. (2019), memory footprint Cai et al. (2018); Marchisio et al. (2020), energy consumption Hsu et al. (2018); Yang et al. (2020) for edge devices. NAS methods are often categorized into three types: gradient-based methods Liu et al. (2018), evolutionary methods Liu et al. (2020), and reinforcement learning (RL)-based methods Jaafra et al. (2019). Among these, gradient-based methods are the most efficient as they can finish searching in only a few hours, compared to thousands of GPU days with other methods.

In this work, to search for a personalized neural architecture for mitigating the data heterogeneity, we adopt an improved variant of the gradient-based method, MiLeNAS He et al. (2020c), which is computationally tractable and particularly suitable for resource-constrained edge devices. Particularly, we propose a new method named Federated NAS (`FedNAS`) to search model architectures among edge devices collaboratively. As shown in Figure 1, `FedNAS` works in the following way. We first utilize the MiLeNAS He et al. (2020c) as a local searcher on each client's local data, which can be distributed easily and efficiently in search time (Step 1). Formally, it formulates NAS as a mixed-level problem: $w = w - \eta_w \nabla_w \mathcal{L}_{\text{tr}}(w, \alpha), \alpha = \alpha - \eta_\alpha \left( \nabla_\alpha \mathcal{L}_{\text{tr}}(w, \alpha) + \lambda \nabla_\alpha \mathcal{L}_{\text{val}}(w, \alpha) \right)$, where $w$ represents the network weight and $\alpha$ represents the neural architecture. $\mathcal{L}_{tr}(w, \alpha)$ and $\mathcal{L}_{val}(w, \alpha)$ denote the loss with respect to training data and validation data, respectively. After the local search, each client then transmits weights $w$ and architecture $\alpha$ to the server (Step 2). The server then applies a weighted aggregation to obtain the server-side $\alpha$ and $w$ (Step 3) and sends the updated parameters back to each client for the next round of searching (Step 4). During the searching process, we can personalize the $\alpha$ and $w$ parameters by alternative local adaptation. Such personalization method can either obtain a higher accuracy for various data distributions, or automate the training process with lightweight hyper-parameter searching efforts.

We evaluate `FedNAS` comprehensively in curated non-I.I.D. datasets, including CIFAR-10 and GLD-23K. Our datasets cover both global model training and personalized model training. We also consider different training scenarios: cross-silo FL and cross-device FL, which has a different total number of clients and number of clients per round. We demonstrate that the personalized model architectures learned by the individual clients perform better than the fine-tuned `FedAvg` and other representative personalized FL methods such `Ditto` Li et al. (2020) and `perFedAvg` Fallah et al. (2020) with default hyper-parameters in most settings.

In summary, our main contributions in this paper are three-fold.

1. We propose the FedNAS method to search for both global model and personalized model architectures collaboratively among edge devices and show its satisfying performance in a variety of FL settings.

2. We investigate the role of NAS to address the challenge of data-heterogeneity in FL and demonstrate via experimental results that it can adapt to users' data better than existing local adaptation and personalization schemes.

3. We experimentally show that FedNAS can achieve state-of-the-art performance for both cross-silo and cross-device settings.

## 2 PROPOSED METHOD

### 2.1 PROBLEM DEFINITION

In the federated learning setting, there are $K$ nodes in the network. Each node has a dataset $\mathcal{D}_k := \left\{ \left( x_i^k, y_i \right) \right\}_{i=1}^{N_k}$ which is non-IID. When collaboratively training a deep neural network (DNN) model with $K$ nodes, the objective function is defined as:

$$\min_w f(w, \underbrace{\alpha}_{fixed}) \overset{\text{def}}{=} \min_w \sum_{k=1}^{K} \frac{N_k}{N} \cdot \frac{1}{N_k} \sum_{i \in \mathcal{D}_k} \ell(x_i, y_i; w, \underbrace{\alpha}_{fixed}) \tag{1}$$

where $w$ represents the network weight, $\alpha$ determines the neural architecture, and $\ell$ is the loss function of the DNN model. To minimize the objective function above, previous works choose a fixed model architecture $\alpha$ then design variant optimization techniques to train the model $w$.

We propose to optimize the federated learning problem from a completely different angle, optimizing $w$ and $\alpha$ simultaneously. Formally, we can reformulate the objective function as:

$$\min_{w,\alpha} f(w, \alpha) \overset{\text{def}}{=} \min_{w,\alpha} \sum_{k=1}^{K} \frac{N_k}{N} \cdot \frac{1}{N_k} \sum_{i \in \mathcal{D}_k} \ell(x_i, y_i; w, \alpha) \tag{2}$$

In other words, for the non-IID dataset scattered across many workers, our goal is to search for an optimal architecture $\alpha$ and related model parameters $w$ to fit the dataset more effectively thus achieve better model performance. In this work, we consider searching for CNN architecture to improve the performance of the image classification task.

### 2.2 SEARCH SPACE

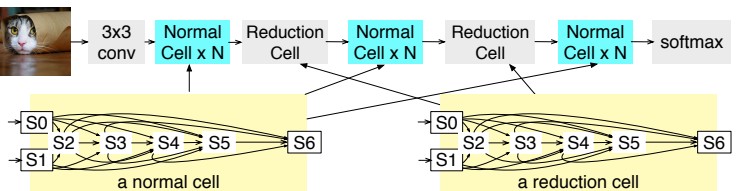

Figure 2: Search Space

Normally, NAS includes three consecutive components: the search space definition, the search algorithm, and the performance estimation strategy Hutter et al. (2019). Our search space follows the mixed-operation search space defined in DARTS Liu et al. (2018) and MiLeNAS He et al. (2020c), where we search in two shared convolutional cells and then build it up as an entire model architecture (as shown in Figure 2). Inside the cell, to relax the categorical candidate operations between two nodes (e.g., convolution, max pooling, skip connection, zero) to a continuous search space, mixed operation using softmax over all possible operations is proposed:

$$\bar{o}^{(i,j)}(x) = \sum_{k=1}^{d} \underbrace{\frac{\exp(\alpha_k^{(i,j)})}{\sum_{k'=1}^{d} \exp(\alpha_{k'}^{(i,j)})}}_{p_k} o_k(x) \tag{3}$$

where the weight $p_k$ of the mixed operation $\bar{o}^{(i,j)}(x)$ for a pair of nodes $(i, j)$ is parameterized by a vector $\alpha^{i,j}$. Thus, all architecture operation options inside a network (model) can be parameterized as $\alpha$. More details are introduced in Appendix A.1.1.

## 2.3 LOCAL SEARCH

Following the above-mentioned search space, each worker searches locally by utilizing the mixed-level optimization technique MiLeNAS He et al. (2020c):

$$
\begin{aligned}
w &= w - \eta_w \nabla_w \mathcal{L}_{\text{tr}}(w, \alpha) \\
\alpha &= \alpha - \eta_\alpha \left( \nabla_\alpha \mathcal{L}_{\text{tr}}(w, \alpha) + \lambda \nabla_\alpha \mathcal{L}_{\text{val}}(w, \alpha) \right)
\end{aligned}
\tag{4}
$$

where $\mathcal{L}_{tr}(w, \alpha)$ and $\mathcal{L}_{val}(w, \alpha)$ denote the loss with respect to the local training data and validation data with $w$ and $\alpha$, respectively.

## 2.4 FEDNAS: FEDERATED NEURAL ARCHITECTURE SEARCH

---
**Algorithm 1** FedNAS Algorithm.

---
1: **Initialization:** initialize $w_0$ and $\alpha_0$; $K$ clients are selected and indexed by $k$; $E$ is the number of local epochs; $T$ is the number of rounds.
2: **Server executes:**
3:    **for** each round $t = 0, 1, 2, ..., T - 1$ **do**
4:       **for** each client $k$ **in parallel do**
5:          $w_{t+1}^k, \alpha_{t+1}^k \leftarrow$ ClientLocalSearch$(k, w_t, \alpha_t)$
6:       $w_{t+1} \leftarrow \sum_{k=1}^{K} \frac{N_k}{N} w_{t+1}^k$
7:       $\alpha_{t+1} \leftarrow \sum_{k=1}^{K} \frac{N_k}{N} \alpha_{t+1}^k$
8:
9: **ClientLocalSearch**$(k, w, \alpha)$: // *Run on client k*
10:    **for** $e$ in epoch **do**
11:       **for** minibatch in training and validation data **do**
12:          Update $w = w - \eta_w \nabla_w \mathcal{L}_{\text{tr}}(w, \alpha)$
13:          Update
14:          $\alpha = \alpha - \eta_\alpha \left( \nabla_\alpha \mathcal{L}_{\text{tr}}(w, \alpha) + \lambda \nabla_\alpha \mathcal{L}_{\text{val}}(w, \alpha) \right)$
15:    return $w, \alpha$ to server

---

We propose FedNAS, a distributed neural architecture search algorithm that aims at optimizing the objective function in Equation 2 under the FL setting. We introduce FedNAS corresponding to four steps in Figure 1: 1) The local searching process: each worker optimizes $\alpha$ and $w$ simultaneously using Eq. 4 for several epochs; 2) All clients send their $\alpha$ and $w$ to the server; 3) The central server aggregates these gradients as follows:

$$
\begin{aligned}
w_{t+1} &\leftarrow \sum_{k=1}^{K} \frac{N_k}{N} w_{t+1}^k \\
\alpha_{t+1} &\leftarrow \sum_{k=1}^{K} \frac{N_k}{N} \alpha_{t+1}^k
\end{aligned}
\tag{5}
$$

4) The server sends back the updated $\alpha$ and $w$ to clients, and each client updates its local $\alpha$ and $w$ accordingly, before running the next round of searching. This process is summarized in Algorithm 1. After searching, an additional evaluation stage is conducted by using a traditional federated optimization method such as *FedAvg* McMahan et al. (2016).

## 2.5 PERSONALIZED FEDNAS: ALTERNATIVE LOCAL ADAPTATION

**Local Adaptation.** To personalize local models, we fine-tune the received global model locally. Such local fine-tuning follows Equation 4, meaning that each client alternatively optimizes its local architecture $\alpha$ and model weights $w$. We found from experiments that such fine-tuning can make a local model more robust against local data heterogeneity, compared with fine-tuning and local adaptation based on *predefined model* and state-of-the-art personalized optimization methods (e.g., Ditto Li et al. (2020) and perFedAvg Fallah et al. (2020)).

**Robust to Varying Data Heterogeneity and Training Scenarios.** Additional to the benefit of data heterogeneity with personalization, an essential feature of FedNAS is that it does not require many

rounds of hyperparameter searching to adapt to diverse data distributions. Most of the time, the default hyper-parameter already perform very well. This property of `FedNAS` is attributed to three aspects described below.

- Intuitively, the personalized architecture and weights have an additive effect in adapting data heterogeneity, compared with solely personalizing the model weight, especially when the architecture search space is huge.

- Most of the personalized methods are built based on an optimization framework with strong prior assumptions, which may not always match the unknown data distribution. For example, Ditto Li et al. (2020) and pFedMe Dinh et al. (2020) utilizes a bi-level optimization and correlate the relationship of aggregated global model and local model by a regularization-based method. Practitioners have to tune the $\lambda$ value to make it work manually. Although perFedAvg Fallah et al. (2020) brings the idea of meta-learning to adapt to data heterogeneity, it is difficult for practitioners to decide the boundary of its meta-train phase and meta-test phase when the data distribution is unknown.

- Different training scenarios also bring additional randomness and uncertainty. For example, in the cross-device setting, the total client number and the client number per round differs from the cross-silo setting significantly, which further increases the difficulty of model selection and hyper-parameter tuning. `FedNAS` may be more resilient to this uncertainty in practice.

To verify the advantage of `FedNAS`, we run experiments to search for both personalized and global models on cross-silo and cross-device settings (see Section 3.1 and 3.2).

## 2.6  AUTOFL SYSTEM DESIGN

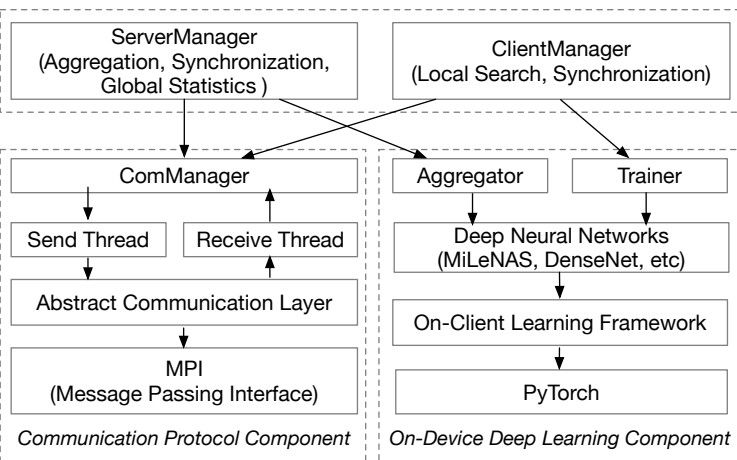

Figure 3: Abstract System Architecture of AutoFL

We design an AutoFL system using FedNAS based on `FedML` He et al. (2020b), an open-source research library for federated learning. The system architecture is shown in Figure 3. This design separates the communication and the model training into two core components shared by the server and clients. The first is the communication protocol component responsible for low-level communication among the server and clients. The second is the on-device deep learning component, which is built based on the popular deep learning framework PyTorch. These two components are encapsulated as *ComManager*, *Trainer*, and *Aggregator*, providing high-level APIs for the above layers. With the help of these APIs, in *ClientManager*, the client can train or search for better architectures and then send its results to the server-side. In contrast, in *ServerManager*, the server can aggregate and synchronize the model architecture and the model parameters with the client-side. More details of the system design can be found in the Appendix.

## 3 EXPERIMENTS AND RESULTS

In this section, we will introduce the experimental results of FedNAS to train a global model as well as personalized models. All our experiments are based on non-IID data distribution among users. In our experiments, we explore two types of non-IID data distributions, label skewed and latent Dirichlet allocation (LDA), which are well explored in literature in FL settings Yurochkin et al. (2019), He et al. (2020a), Arivazhagan et al. (2019).

**Implementation and Deployment.** We set up our experiment in a distributed computing network equipped with GPUs. We perform experiments for two settings, FedNAS for a global model search and FedNAS for personalized models search. For investigating the former setting, we set up our experiment in a cross-silo setting for simplicity and use 17 nodes in total, one representing the server-side and the other 16 nodes representing clients, which can be organizations in the real world (e.g., hospitals and clinics). For personalized model search, we use a larger set of nodes, 21, in total, one representing the server-side and the other 20 nodes representing clients. We pick four clients at random for each round of FedNAS. For all these experiments, each node is a physical server that has an NVIDIA RTX 2080Ti GPU card inside. We deployed the FedNAS system described in Appendix 2.6 on each node. Our code implementation is based on PyTorch 1.4.0, MPI4Py [1] 3.0.3 , and Python 3.7.4.

**Task and Dataset.** Our training task is image classification on the CIFAR10 dataset, which consists of 60000 32x32 color images in 10 classes, with 6000 images per class. For global model searching via FedNAS, we generate non-IID (non identical and independent distribution) local data by splitting the 50000 training images into $K$ clients in an unbalanced manner: sampling $\mathbf{p}_c \sim \mathrm{Dir}_J(0.5)$ and allocating a $\mathbf{p}_{c,k}$ proportion of the training samples of class $c$ to local client $k$. The 10000 test images are used for a global test after the aggregation of each round. The actual data distribution used for this experiment is given in Table 3.

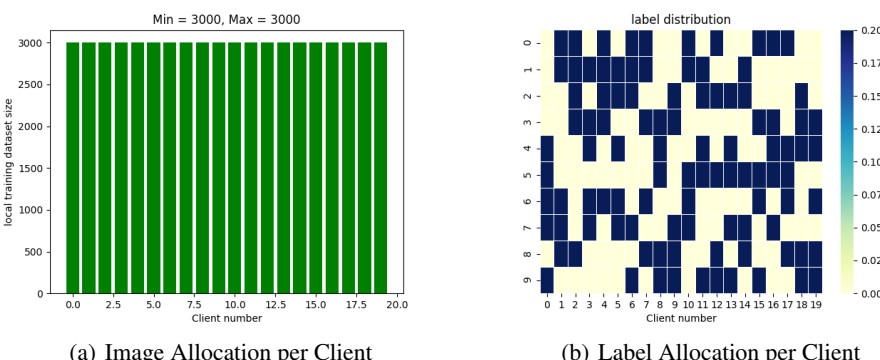

(a) Image Allocation per Client

(b) Label Allocation per Client

Figure 4: CIFAR10: Label Skew Partition

For personalized model experiments, we generate non-IID data by label skewed partition. In this partition scheme, we assign images of only five classes to each client and keep the number of images per client the same, namely 3000, as shown in Figure 4. For each client, we further split these 3000 images into the training and testing datasets by using 75% data, i.e., 2250 images, for training, and the other 25% as testing data. We perform this split to test personalization as it requires each client to have their own local test dataset. We also explore latent Dirichlet distribution (LDA) based non-IID data distribution for the personalized model setup, and details of this distribution for this experiment can be found in the appendix A.1.2. Since the model performance is sensitive to the data distribution, we fix the non-IID dataset in all experiments for a fair comparison.

---

[1] https://pypi.org/project/mpi4py/

### 3.1 PERSONALIZED MODELS SEARCH VIA FEDNAS

To demonstrate the efficacy of FedNAS to design better local models, we compare FedNAS with local adaption (via FedAvg), Ditto and perFedAvg. Besides FedNAS, every other method runs on a manually designed architecture, ResNet18 Targ et al. (2016), which has more model parameters, 11M, than the 8-layer DARTs cell structure of FedNAS which has only 4M model parameters He et al. (2020c). To evaluate the performance, we use average validation accuracy of all clients as a performance metric.

#### 3.1.1 RESULTS ON NON-I.I.D. (LABEL SKEW PARTITION AND LDA DISTRIBUTION)

Table 1 illustrates the performance comparison of `FedNAS` with `local adaptation`, `Ditto` and `perFedAvg`. For a fair comparison, we fine-tune hyper-parameters of each method, such as we fine-tune learning rate (lr) hyperparameter over the set {0.1, 0.3, 0.01, 0.03, 0.001, 0.003} of each method. Batch size has been fixed to 32 for all these comparisons. For `Ditto`, in addition to lr, we tune $\lambda$ over the set of {2, 1, 0.1, 0.01, 0.001}. For perFedAvg, we tune the global lr over {0.1, 0.3, 0.01, 0.03, 0.001, 0.003} by keeping the local lr {1, 3, 5, 7, 10} times higher than the global lr.

Table 1 draws the comparison between different methods for the average validation accuracy of all the clients metric for lda and label skew distribution. Interestingly, FedNAS outperforms all other methods for both label skew and lda distribution which highlights its power to adapt to user's data well and perform better locally as well. Overall, for label skew distribution, it achieves average validation accuracy of 91.3% which is 5% higher than the local adaptation's validation accuracy and 2% higher than `Ditto`. We also observe that `Ditto` outperforms the local adaptaion in terms of validation accuracy but have higher standard deviation than local adaptation.

Table 1: Average local validation Accuracy Comparison of FedNAS with other personalization techniques)

| Method | Parameter size | Accuracy (Label Skew) | Accuracy (lda Distribution) |
|--------|----------------|----------------------|------------------------------|
| **FedNAS** | 4M | 0.913±0.025 | 0.907±0.024 |
| Local Adaptation | 11M | 0.864±0.028 | 0.861±0.0357 |
| Ditto | 11M | 0.894±0.035 | 0.88 ±0.032 |
| perFedAvg | 11M | 0.888±0.036 | 0.894±0.032 |

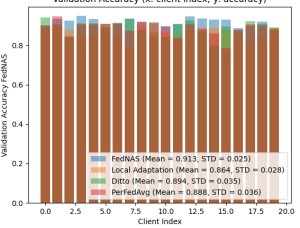
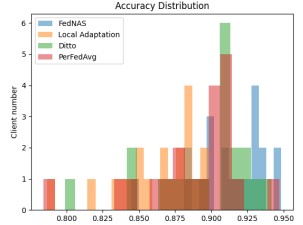
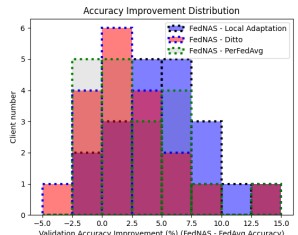

(a) Local Validation Accuracy    (b) Average Validation Accuracy Distribution    (c) Average Validation Accuracy Improvement

Figure 5: Visualization of validation accuracy of each client and accuracy improvement distribution for personalized model search.

For detailed comparison, we compare the validation accuracies of all the clients for the best round of label skew distribution. The best round for each method is selected as the round that provides the highest average validation accuracy. We visualize the validation accuracy of each client 5(a), average validation accuracy distribution 5(b) and average validation improvement distribution 5(c). For accuracy improvement distribution, we subtract the validation accuracy of `FedNAS` from the respective method for each client and plot the histogram. The improvement histogram shows that for

one of the clients the improvement can be as high as 15% as compare to perFedAvg. As compare to `Ditto`, we get 2.5% improvement for even 6 clients. On the other hand, there are only two clients, client number 10 and 20, for which local adaptation performs slightly (2.5%) better. Although there are some clients for which FedNAS do not perform well as compared to these methods, it is important to note that the standard deviation of FedNAS more prominent in figure (b) is lowest and accuracy histograms are concentrated towards the right side, whereas for other methods, these bars fall as below as 82% accuracy.

## 3.2 GLOBAL MODEL SEARCH VIA FEDNAS

To investigate the performance of `FedNAS` to design a global model, we search a global model via FedNAS and compare it to the well-known FL algorithm `FedAvg`, which runs on DenseNet Huang et al. (2017), a manually designed architecture that extends ResNet He et al. (2016) but with higher performance and fewer model parameters. We run both of these experiments on the same non-IID dataset.

### 3.2.1 RESULTS ON NON-I.I.D. (LDA PARTITION)

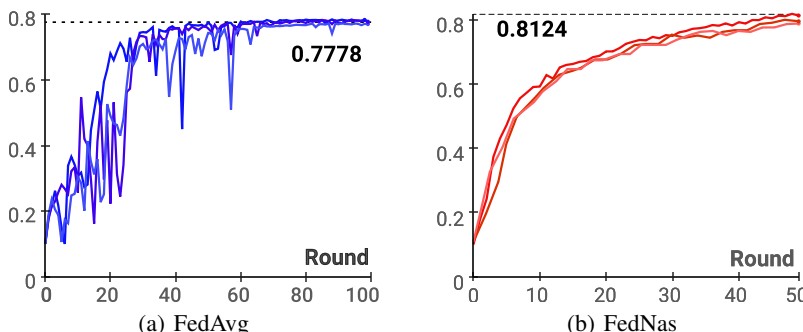

Figure 6: Test Accuracy on Non-IID Dataset (multiple runs) FedAvg on DenseNet vs. FedNAS

Figure 6 demonstrates the performance of FedNAS vs. FedAvg. We use a specific non-IID data distribution given in appendix A.1.2 and keep it fixed for both experiments. For a fair comparison, results are obtained by fine-tuning hyperparameters of each method, and each method is run three times. Details of hyperparameter tunning can be found in the appendix A.1.4.

Figure 6(a) shows the global test accuracy during the training process of FedAvg, whereas, Figure 6(b) reports the global test accuracy during the searching process of `FedNAS`. Global test accuracy has been calculated using the 10000 test images of CIFAR10 dataset. First of all, we demonstrate the compatibility of NAS for the data-heterogeneous FL setting. In addition to convergence of `FedNAS`, we show that `FedNAS` can achieve higher accuracy than FedAvg during the searching process (81.24% in Figure 6(b); 77.78% in Figure 6(a)). This 4% performance benefit further confirms the efficacy of `FedNAS`. We also evaluate the searched architecture under this data distribution. We found that each run of `FedNAS` can obtain a higher test accuracy than each run of `FedAvg`. On average, the architecture searched by `FedNAS` obtains a test accuracy 4% higher than `FedAvg`.

Hyperparameters and visualization of the searched architecture can be found in Appendix A.1.4 and A.1.5, respectively.

**Remark**. *We also ran experiments on other distributions of non-IID datasets, in which FedNAS is also demonstrated to beat* `FedAvg`, *confirming that* `FedNAS` *searches for better architectures with a higher model performance.*

## 3.3 EVALUATION OF THE SYSTEM EFFICIENCY

In order to more comprehensively reflect our distributed search overhead, we developed the single-process and distributed version of `FedNAS` and `FedAvg`. The single-process version simulates the algorithm by performing a client-by-client search on a single GPU card. As shown in Table 2,

Table 2: Efficiency Comparison (16 RTX2080Ti GPUs as clients, and 1 RTX2080Ti as server)

| Method | Search Time | Parameter Size | Hyperparameter |
|---|---|---|---|
| FedAvg (single) | > 3 days | - | rounds = 100, local epochs=20, batch size=64 |
| FedAvg (distributed) | 12 hours | 20.01M | |
| FedNAS (single) | 33 hours | - | rounds = 50, local epochs=5, batch size=64 |
| **FedNAS (distributed)** | **< 5 hours** | **1.93M** | |

compared with `FedAvg` and manually designed DenseNet, `FedNAS` can find better architecture with fewer parameters in less time. `FedAvg` spends more time because it requires more local epochs to converge.

## 4 RELATED WORKS

Recently, Neural Architecture Search (NAS) Hutter et al. (2019) has attracted widespread attention due to its advantages over manually designed models. There are three major NAS methods: evolutionary algorithms, reinforcement learning-based methods, and gradient-based methods He et al. (2020c). While in the Federated Learning (FL) domain McMahan et al. (2016); He et al. (2019), using pre-designed model architectures and optimizing by FedAvg McMahan et al. (2016) is the main method to improve model performance. To our knowledge, NAS is rarely studied in FL setting to study the aspect of personal model search for real-time setting. Although Kairouz et al. (2019) first proposed the concept of automating FL via NAS, the concrete method and details are never given. There are a few works done in the direction of the use of a NAS to search for a global model, however, no personalization exploration is provided.

For global model search, Zhu & Jin (2020) is a NAS based FL work that exploits evolutionary NAS to design a master model but they utilize double-client sampling to make their method edge resource friendly. Contrary to this, we exploit gradient-based NAS method, `MileNAS`, which is comparatively faster and more resource friendly than the evolutionary and reinforcement-based methods. The other work in this direction is Singh et al. (2020) that explores the concept of differential privacy by using DARTs as a NAS solver to search for a global model. However, our proposed work uses MileNAS solver which has an extensive analysis of its performance efficiency over DARTS given in the original MileNAS work He et al. (2020c). Another work Garg et al. (2020) uses DSNAS which is another gradient based NAS algorithm to search for a global model. DSNAS works on sampling a child network from a supernetwork even in search phase whereas MileNAS solver searches over the complete supernetwork, and therefore, has the potential to provide more freedom to clients to search for a better and personalized architecture. Another work Xu et al. (2020) proposes a very different idea than conventional neural architecture search where they begin with a pretrained manually designed model and keep pruning the model until it satisfies the efficiency budget. Although they named their work as federated neural architecture search but model search is performed on the server side alone, none of the clients participate in searching a better model (finding architecture parameters). Clients only participate in training the pruned model's parameters communicated to clients by server. Furthermore, to the best of our knowledge, we are the first work that investigate the performance of locally searched architectures in federated NAS.

## 5 CONCLUSION

In this paper, we propose FedNAS, a unified neural architecture search based federated learning framework to design a global model collaboratively. First, we study the compliance of gradient based neural architecture search algorithm, MileNAS, with the FedAvg algorithm. We analyze its performance for both cross silo and cross-device settings, and show its convergence for both setups. We also investigate the proposed framework, FedNAS, from the perspective of personalization and its role to overcome the challenge of data-heterogeneity in FL. To test data-heterogeneity, we explore FedNAS for both label-skewed and lda-based non-IID data distributions, and show via experimental results its superiority over other personalization methods, such as local fine-tunning, `Ditto` and `perFedAvg`.

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
