# OpenReview forum: "FedNAS: Federated Deep Learning via Neural Architecture Search"
_ICLR.cc/2022/Conference — ICLR 2022 Submitted_

### Official Review · Reviewer_xdiq · 2021-10-31

**Correctness:** 4
**Technical Novelty And Significance:** 2
**Empirical Novelty And Significance:** 3
**Recommendation:** 6
**Confidence:** 3

**Main Review:**

The global model search with FedNAS is better is not surprising, since there is no difference between training on FL and centralized settings, on only gradients. What becomes really interesting, is the personalization performance. Comparing to existing personalization methods which fix network structure and personalize weights only, FedNAS optimizes in both neural architecture and weights, which results in better personalized performance (both by average and by individual), and smaller model.

Contributions:

The paper's work is a natural extension of MileNAS. Since MileNAS, like DARTS, is gradient-based algorithm, then make it available to Federated Learning is natural. In this sense, the innovation is not extraordinary.

There are some minors to be improved:

(1) Experiments are not sufficient. Although for CIFAR10 and GLD-23k, the results are pretty extensive, more experiments on other FL datasets (EMNIST-62, CIFAR100, and even Shakespeare), could make the paper stronger.

(2) Moreover, FedNAS search architecture only applies for Computer Vision (specifically, image classification problems). How to extend NAS to NLP tasks or other paradigms, could be an interesting next step.

(3) Interpretation of the learned architecture. The performance gain originates from personalize both architecture and weights. The author confirms that the personalized performance does improve, but the interpretation is missing. It could be interesting, to see what if some clients learns less/more complex models.


**Summary Of The Paper:**

This paper combines gradient-based NAS (DARTS-like algorithm: MileNAS) with Federated Learning (FL) setup, to improve both global and personalization performance with learned neural architecture. Since both NAS and FL learning are based on gradient, the extension to FL setup becomes intuitive and effective. Empirically FedNAS shows improved performance comparing to existing FL methods.

**Summary Of The Review:**

Overall, the proposed method to use gradient-based NAS for FL to improve personalization, shows good empirical performance. The complaint of mine, is that the innovation is on applying existing method (MileNAS, or DARTS-like gradient-based NAS), directly onto FL settings. There are very few modification on original method, other than the settings are different.

In all, I give this paper score 6, and I am willing to discuss with other reviewers and the authors in later rebuttal session.

---

> ### Author Response · Authors · 2021-11-23
> **Author Response**
>
> Thanks for your constructive feedback.
>
> * ***Q1*** Experiments are not sufficient. Although for CIFAR10 and GLD-23k, the results are pretty extensive, more experiments on other FL datasets (EMNIST-62, CIFAR100, and even Shakespeare), could make the paper stronger.
>
> ***Response***: GLD-23k is the representative dataset suggested by Google FL team. It's high-resolution CV dataset, compared to EMNIST-62 and CIFAR-100. As such, we believe our experiment is adequate.
>
> * ***Q2***: Moreover, FedNAS search architecture only applies for Computer Vision (specifically, image classification problems). How to extend NAS to NLP tasks or other paradigms, could be an interesting next step.
>
> ***Response***: Given that NLP is mainly Transformer model-based and methods of NAS for Transformer are very limited, it's not very meaningful to further extend FedNAS to NLP domain. But indeed, you suggest a very promising direction. We will try to figure out whether there is practical solution for Transformer-based FedNAS.
>
>
> * ***Q3***: Interpretation of the learned architecture.
>
> ***Response***: Thank you for your suggestion. You can see that Table 1 already demonstrates the searched architecture is sparse. We will add more visualization to understand the structure.
>
> Overall, thanks for your suggestions. It's highly appreciated if you could further guide the discussion and recommend other reviewers to reconsider the rating. Thanks.

---

### Official Review · Reviewer_3t92 · 2021-11-01

**Correctness:** 3
**Technical Novelty And Significance:** 2
**Empirical Novelty And Significance:** 2
**Recommendation:** 5
**Confidence:** 4

**Main Review:**

Strength:
1. The paper is generally clearly written.
2. The codes are open-sourced for reproducibility.

Weakness:
1. The motivation is not clear. The abstract says "We propose Federated Neural Architecture Search (FedNAS) for automating the model design process in FL." and the introduction says "We aim to address data heterogeneity in FL via ... NAS." For me, it is not clear whether the paper aims to automate model design in FL or solve the non-IID problem.
2. NAS method and FL are naively combined, which undermines the novelty of this paper.
The comparison experiments conducted might be not fair enough. Model searched by NAS are generally smaller than handcrafted ones while enjoying higher accuracy. For example, in [1], a 4.6M ENAS model outperforms a 25.6M DenseNet model. In comparison, I think a DARTS model should be used as a base for previous FL methods.

[1] H.Pham et.al, Efficient Neural Architecture Search via Parameter Sharing. ICML’18.

**Summary Of The Paper:**

This paper proposes a method for automatic model design in the context of federated learning and introduces an auto-ml system under the framework of federated learning.



**Summary Of The Review:**

None

---

> ### Author Response · Authors · 2021-11-23
> **Author Response**
>
> * ***Q1*** The motivation is not clear. The abstract says "We propose Federated Neural Architecture Search (FedNAS) for automating the model design process in FL." and the introduction says "We aim to address data heterogeneity in FL via ... NAS." For me, it is not clear whether the paper aims to automate model design in FL or solve the non-IID problem.
>
> ***Response***: We will refine the expression in our revision. In fact, we want to highlight that FedNAS aims to search for personalized architecture automatically under non-IID data. (invisible to the model developer at the server-side)
>
>
> * ***Q2*** NAS method and FL are naively combined, which undermines the novelty of this paper. The comparison experiments conducted might be not fair enough. Model searched by NAS are generally smaller than handcrafted ones while enjoying higher accuracy. For example, in [1], a 4.6M ENAS model outperforms a 25.6M DenseNet model. In comparison, I think a DARTS model should be used as a base for previous FL methods.
>
> ***Response***: we have to highlight that having the idea of integrating NAS to FL is novel (we are the first paper to propose this idea). Additionally, we contribute
> 1) a simple yet effective way to personalize the model architecture by fine-tuning after a global searching.
> 2) carefully compare personalized FL method and demonstrate our FedNAS can beat them automatically without intensive hyperparameter search.
> 3) Given the simple yet effective of our method, we further build a strong distributed training system based on FedNAS. This will serve as a strong baseline for future research.
>
> As for the ENAS baseline, it's already been verfied in centralized NAS that ENAS is worse than gradient-based NAS such as DARTS and MiLeNAS. After a careful comparison, for federated learning, resource and robust-efficient method like MiLeNAS is our first choice. We have to highlight one philosophy that "if there is clear conclusion in centralized ML that method A works better than method B in terms of efficiency and accuracy, there is no need to further compare it in distributed/federated setting." Research progress builds on previous results, if we brings too many redundant baselines to FL, it will slows the research progress. Hope you can understand this perspective.
>
>
> Overall, we think these two minor issues are not strong reasons to reject our work. It's highly appreciated if you could further reconsider the rating. Thanks.

---

> > ### Comment · Reviewer_3t92 · 2021-11-29
> > **The point of Question 2**
> >
> > I would like to clarify the point of Question 2.
> >
> > ENAS is mentioned to provide evidence showing that: "Model searched by NAS are generally smaller than handcrafted ones while enjoying higher accuracy". So I am not sure about the fairness of comparing FedNAS, which utilizes NAS, against FedAvg with a handcrafted model. I think a NAS-based model is supposed to be used for FedAvg. Comparing the FedNAS and ENAS is not the point.
> >
> >  I am happy to raise my score if this problem is addressed.

---

> > > ### Author Response · Authors · 2021-11-30
> > > **Additional experimental result**
> > >
> > > We thank the reviewer for the insightful comment. We tried experiments today. According to our exploration, the accuracy obtained by FedNAS with personalized architecture can still beat the baseline you suggested (FedAvg + the neural architecture found by the DARTS paper). This is reasonable since the baseline trains a sparse architecture from scratch, while our FedNAS with personalization is a dynamical learning process with alternating minimization and adapting to local heterogeneous data distribution.
> > >
> > > We also double check this by replacing the DARTS model with the NAS searched model in FL setting (architecture given in Figure 10 and 11 in Appendix), FedNAS still beats it. The baseline's average validation accuracy value (averaged across all clients) is quite low, i.e., 76%, which is 14 % lower than our proposed method. The hyperparameters are provided below.
> > >
> > > We will further clarify this in our revision. Please help to raise the score, since there is only 1 hour left.
> > >
> > > ----
> > > Experiment Hyperparameters: we kept batch size to be 32, local epoch 1 and 1000 communication rounds. For lr, we searched over the set {0.001,0.01, 0.1, 0.3, 0.03, 0.003}.  We used the same lda based distribution with CIFAR10 dataset as is used in our reported experiments (lda distribution illustrated in Figure 7 in Appendix).

---

> > > > ### Author Response · Authors · 2021-11-30
> > > > **Additional experimental result**
> > > >
> > > > Here are the final accuracy values obtained from our exploration.
> > > >
> > > >
> > > > | Method               | Average Validation Accuracy |
> > > > |----------------------|-----------------------------|
> > > > | FedNAS               | 90.64%                      |
> > > > | FedAvg + DARTs model | 87.11%                      |
> > > >
> > > > Hyper-parameter update: for FedAvg + DARTS model, we used 3500 communication rounds until the model converges to report average validation accuracy across all clients.

---

### Official Review · Reviewer_y4xh · 2021-11-02

**Correctness:** 3
**Technical Novelty And Significance:** 3
**Empirical Novelty And Significance:** 3
**Recommendation:** 5
**Confidence:** 4

**Main Review:**

Strengths:
1. The authors address the data heterogeneity problems in federated learning via NAS methods. The personalized architecture shows good improvements.


Weakness:
1. Originality: This work is a simple extension of MiLeNAS on federated learning. The NAS on clients for personalization seems different from other NAS works. But I did not see any challenge highlighted by authors and they directly adopt NAS methods for common scenarios.
2. Quality: The aggregation on the central server follows the similar way of FedAvg. The most difference with other existing works is to introduce NAS into federated learning. However, during employing NAS into federated learning, only trivial changes are made. When architecture changes, the global aggregation still remains the similar way in traditional federated learning and leaves some unexplored spaces. Even though MiLeNAS is a recent effective method, other NAS methods should be introduced into federated learning to justify the authors' choice.
3. Soundness: One of the most important parts of federated learning especially in new settings is convergence guarantee. The architectures on clients keep changing and personalization is needed, resulting in more heterogeneous model weights and architectures. How to guarantee the convergence of the proposed framework? I will suggest authors provide some theoretical proof to support this method for convergence.

**Summary Of The Paper:**

The authors employ an existing neural architecture search method in the federated learning setting.  Specifically, the authors propose FeNAS and extend an existing NAS method MiLeNAS into federated learning to address the data heterogeneity problem and conduct personalization.  The experiments show that the proposed method is able to achieve improvement compared to some other federated learning methods.

**Summary Of The Review:**

The authors extend an existing NAS method MiLeNAS on federated personalization. However, the novelty of the work is somewhat limited.  It is lacking clear support evidence and insights to support why MiLeNAS was chosen. To support the idea of NAS for federated learning, more other NAS methods are preferred to provide more insights and depict the challenges of this problem.

---

> ### Author Response · Authors · 2021-11-23
> **Author Response**
>
> * ***Q1***: Originality
>
> Response: We have to highlight that having the idea of integrating NAS to FL is novel (we are the first paper to propose this idea). Additionally, we contribute
> 1) a simple yet effective way to personalize the model architecture by fine-tuning after a global searching.
> 2) carefully compare personalized FL method and demonstrate our FedNAS can beat them automatically without intensive hyperparameter search.
> 3) Given the simple yet effective of our method, we further build a strong distributed training system based on FedNAS. This will serve as a strong baseline for future research.
>
> * ***Q2***: Quality: no special aggregation.
>
> Response: we have to highlight that our FedNAS doesn't use heterogeneous architecture during the first searching phase, i.e., the alpha architecture learnable parameters are probability of each operation in neural networks, meaning that naive weighted aggregation is enough to do aggregation for alpha parameters. Later, after searching, we use fine-tuning for local alpha variable.
>
> Such local fine-tuning follows Equation 4, meaning that each client alternatively optimizes its local architecture α and model weights w. We found from experiments that such fine-tuning can make a local model more robust against local data heterogeneity, compared with fine-tuning and local adaptation based on predefined model and state-of-the-art personalized optimization methods (Ditto Li et al. (2020) and perFedAvg (Fallah et al. (2020)).
>
> * ***Q3***: Convergence Analysis
> Given the lack of theory in NAS domain (e.g., even for the NAS-related outstanding paper at ICLR 2021, there is no theory), we think it's hard for us to provide analysis in FL setting. Do you have any suggestions?
>
> Overall, we think these three concerns are not strong reasons to reject our work. It's highly appreciated if you could further reconsider the rating. Thanks.

---

### Official Review · Reviewer_sG5T · 2021-11-04

**Correctness:** 3
**Technical Novelty And Significance:** 3
**Empirical Novelty And Significance:** 2
**Recommendation:** 5
**Confidence:** 3

**Main Review:**

Strengths:
1. The paper propose the Federated Neural Architecture Search (FedNAS) method to search for both global model and personalized model architectures collaboratively among edge devices, and show its performance in a variety of federated learning  settings.
2. It also investigate the role of NAS to address the challenge of data-heterogeneity in federated learning and show superiority in results as well as its efficacy.


Weakness:
1. It was not clear the purpose of Figure 4 (1) given that the graph shows uniform local training dataset size.
2. It seems that the paper is missing some strong baseline in NAS spaces. In related works, authors lists our some work in NAS, such as Kairouz et al,, 2019, Zhu & Jin 2020, etc.. The paper would be stronger to also benchmark against related methods in NAS.

**Summary Of The Paper:**

The paper propose the Federated Neural Architecture Search (FedNAS) method to search for both global model and personalized model architectures collaboratively among edge devices and show its performance in a variety of federated learning  settings. It also investigate the role of NAS to address the challenge of data-heterogeneity in federated learning and show superiority in results.

**Summary Of The Review:**

Overall technical seems significant and somewhat new, but as mentioned above in the weakness, the paper is missing some comparison against stronger baselines, especially for previous methods in NAS.

---

> ### Author Response · Authors · 2021-11-23
> **Author Response**
>
> * ***Q1***: It was not clear the purpose of Figure 4 (1) given that the graph shows uniform local training dataset size.
>
> ***Response***: Figure 4 is for "CIFAR10: Label Skew Partition", meaning that each client only has 5 labels, though they have the same number of samples.
>
>
> * ***Q2***: It seems that the paper is missing some strong baseline in NAS spaces. In related works, authors lists our some work in NAS, such as Kairouz et al,, 2019, Zhu & Jin 2020, etc.. The paper would be stronger to also benchmark against related methods in NAS.
>
> ***Response***: (Zhu & Jin 2020) is an evolutionary-based method. In centralized NAS, researchers have already demonstrated gradient-based NAS (DARTS and MiLeNAS) is comparatively faster and more resource friendly than the evolutionary and reinforcement-based methods. So it's clear that our method fits more for FL setting.
>
> But we do agree that comparing related works is always helpful. To compare with (Zhu & Jin 2020), we have to get the original source code. However, after checking this link: https://paperswithcode.com/paper/real-time-federated-evolutionary-neural, we cannot find any related source code to help us to reproduce the work. We may need to contact the author later to reproduce their results. Thanks for your suggestions.
>
> Overall, we think these two minor issues are not strong reasons to reject our work. It's highly appreciated if you could further reconsider the rating. Thanks.

---

### Comment · Area_Chair_k5za · 2021-11-28
**Please provide feedback**

Dear Reviewers,

The discussion deadline is only two days away, and we need to reach a consensus soon to recommend a decision. Please go over the responses from the authors on your comments, and revise your ratings or reviews when necessary.

Thanks, Area Chair

---

### Decision · Program_Chairs · 2022-01-20

**Decision:**

Reject

**Comment:**

This paper proposes a personalized federated learning framework based on neural architecture search, in which the local clients perform NAS to search for a better architecture for the private local data. Specifically, the authors extend MiLeNAS, which is an existing NAS algorithm, to be run in the federated learning setting, and use FedAvg for model aggregation. The proposed FedNAS framework is validated against personalized federated learning methods with predefined architectures, such as perFedAvg, Ditto, and local fine-tuning, and is shown to largely outperform them on non-IID settings with label skew and LDA distribution. FedNAS’s collaborative search for the optimal architecture also yields a better performing global model than FedAvg.

The paper received borderline ratings. Three out of four reviewers are learning negative, while one is leaning negative. The below is the summary of pros and cons of the paper mentioned by the reviewers:

Pros
- The idea of using NAS for personalized federated learning seems novel and interesting.
- The proposed FedNAS framework is shown to be effective in tackling the data heterogeneity problem, which is a fundamental problem with federated learning.
- The authors have released the code for reproducibility.

Cons
- The technical contribution of the work seems limited, since the proposed FedNAS straightforwardly combines an existing NAS method (MiLeNAS) with federated averaging, and there is no challenge mentioned for this new problem of federated NAS.
- The choice of a specific NAS method (MiLeNAS) is not well justified, and other NAS methods should be also considered.
- The motivation is unclear: It is not clear whether the authors aim to perform collaborative automotive design or solve personalized federated learning.
- There is no convergence analysis.

While some of the concerns have been addressed away in the authors’ responses during the rebuttal period, the reviewers did not change their ratings, and the final consensus was to reject the paper.

I agree with the authors that combining federated learning with NAS, and applying it for personalized federated learning is a novel idea that intuitively makes sense. However, I agree with the reviewers that the current method is a straightforward combination of an existing NAS method and an existing FL algorithm, the authors should identify new challenges posed by the combination of the two methods, and identify them.

Further, performing NAS on edge devices may be possible, but not the best solution, since it could result in large computational overhead. While the authors mention that MiLeNAS is computational suitable in such settings, there should be a proper investigation of the accuracy-efficiency tradeoff, showing how well FedNAS performs against others with the same computational budget (or training time / energy consumption).

Overall, this is a paper that proposes a novel and interesting idea that seems to work, but the paper does not sufficiently examine challenges posed by the new problem. I suggest the authors identify the new challenges and examine the efficiency issue mentioned, and further develop their method, if necessary.